REGISTERED REPORT PROTOCOL

# Examining the meaning and methodological characteristics of the systematized review label: A scoping review protocol

Zahra Premji [1], Leyla Cabugos [2] *

1 Advanced Research Services-Libraries, University of Victoria, Victoria, British Columbia, Canada,
2 Academic Services, Robert E. Kennedy Library, California Polytechnic State University, San Luis Obispo, California, United States of America

☯ These authors contributed equally to this work.
* lcabugos@calpoly.edu

## Abstract

### Introduction

A large portion of published evidence syntheses (ES) do not conform to established standards. There is a growing number of reviews labeled as a "systematized review", but authors do not always identify specific methodological adaptations or specify the biases these may introduce.

### Objective

The objective of this scoping review is to identify which methodological attributes common to evidence syntheses (ES) are implemented or referenced in published systematized reviews. It also aims to 1) identify and collate, where available, the reasons authors characterize their study as a systematized review 2) determine whether any justifications provided were based on resource constraints or research goals, and 3) describe common characteristics of systematized reviews.

### Inclusion criteria

All articles that are titled as a systematized review, attempt a collocation and synthesis of existing literature, and include some methodology for their review process, will be included.

### Materials and methods

A title search will be conducted for the terms "systematized" or "systematised" in proximity to the term "review" in a selection of scholarly sources that offer broad coverage of literature in many disciplines: Google Scholar, Lens, Web of Science Core Collection (Web of Science platform), Scopus (Elsevier platform), MEDLINE (Ovid platform). Screening and data extraction will be done in duplicate. Screening will be conducted in Covidence. Data extraction will be done in Google sheets. Data extraction elements will include common methodological characteristics relating to various steps of the evidence synthesis process.

This is a Registered Report and may have an associated publication; please check the article page on the journal site for any related articles.

**Data Availability Statement:** All relevant data from this study will be made available upon study completion.

**Funding:** Publishing fees for this work are provided by an internal (RSCA) grant secured by the corresponding author. RSCA grant recipients must use the following language when recognizing support of their RSCA project: "This material is based upon work supported by the Research, Scholarly & Creative Activities Program awarded by the Cal Poly division of Research, Economic Development & Graduate Education." There is no grant number associated with the award.

**Competing interests:** The authors have declared that no competing interests exist.

Descriptive, aggregate statistics, and categorization of reasons for selecting the systematized review type are the primary planned analysis for this review.

## Data availability

This is a registered report protocol. The data collected in this research project will be made available in the Borealis repository (https://borealisdata.ca/) upon finalization of the study.

## Introduction

Evidence synthesis is the process and product of comprehensively reviewing "what is known from existing research using systematic and explicit methods in order to clarify the evidence base." [1, p.2]. A growing number of study types falls under this umbrella, all designed to produce a picture of the state of evidence, while accounting for sources of bias in research conduct (such as exclusion of relevant variables in participant selection), publishing (such as the over-representation of positive results), and review practices themselves (such as the arbitrary exclusion of evidence in particular languages, geographical regions, or publication venues). Some types of evidence synthesis, such as the systematic review, are associated with standards for conduct and reporting that have been developed and vetted by communities of practice and organizations such as the Campbell Collaboration, Cochrane, the Center for Environmental Evidence, and JBI. Researchers, practitioners, policymakers and funders rely upon evidence syntheses to keep abreast of the voluminous primary research literature [2]. Likely due to their consequential role in decision making, their prestige as the gold standard of evidence, and growing interest in open research practices, the volume and disciplinary reach of evidence syntheses are growing rapidly.

Producing a methodologically rigorous evidence synthesis takes substantial time, and requires multiple competencies (information retrieval, subject knowledge, research design, and sometimes statistics), more than one author, and comprehensive access to literature and tools for data management. A large portion of published evidence syntheses do not conform to established standards [3], leading to the development of frameworks for evaluating their quality [4]. There is a growing body of literature on the quality and reporting issues with published evidence syntheses. A recent living systematic review aimed to collate and gather on an ongoing basis the evidence indicating these issues [5]. While many such departures go unremarked, there seems to be an emerging practice of labeling a study a "systematized review" when it does not quite meet the standards of the systematic review, a type of evidence synthesis with a known methodology. In possible contrast to review types, such as rapid reviews, that offer somewhat codified options for attenuating full evidence synthesis methodology to deliver timely syntheses in the face of resource constraints [6], methodological adaptations in published systematized reviews are varied, and may not be explicitly identified by the authors. A 2009 typology of evidence syntheses characterized the systematized review as an "attempt to include elements of systematic review process while stopping short of claiming that the output is a systematic review" [6, p.102]. As noted in the typology, examples of deviation from the systematic review process could include a single database search followed by a more rigorous analysis of the included studies, or alternatively a full comprehensive search followed by a simple presentation of the results. The typology further characterizes systematized reviews primarily as a vehicle for graduate students to "demonstrate an awareness of the entire process and technical proficiency in the component steps" [p.103] while working under resource

constraints. However, we observe that systematized reviews are published by authors at various career stages.

As we have observed authors modeling their research methods on published examples, and citing relevant guidelines infrequently [7], we expect the prevalence of published systematized reviews to increase. We believe it is helpful to understand what researchers mean, and model for each other, when they use this label. Therefore, this scoping review seeks to identify which elements of evidence synthesis methodology are being included, and which, if any, guidelines are invoked and implemented in systematized reviews. A preliminary search for "systematized" (OR "systematised") AND "review" in the Open Science Framework, Google Scholar, and Library, Information Science and Technology Abstracts, was conducted on 1/10/2023 and no published or planned systematic reviews or scoping reviews on the topic were identified.

The exploratory nature of our research questions make a scoping review an appropriate method for this research. And even though we will be evaluating specific elements of the included systematized reviews (using simple counts, presence or absence of a given element, etc), we will be stopping short of fully assessing each of the included reviews. Furthermore, our objective of mapping the breadth and extent of this type of review in the literature, collating the reasons that authors apply this label, and describing the common methodological characteristics of currently published systematized reviews is in alignment with the reasons for conducting a scoping review [8].

## Review question

The purpose of this research is to describe the current extent and state of systematized reviews and to describe their methodological characteristics. Additionally, this study will seek to identify and collate the reasons researchers provide for selecting the suite of methods they present as a systematized review. Recommendations for future researchers who intend to conduct a systematized review will also be presented.

Research methods evaluation; Evidence synthesis; Methodological review.

## Eligibility criteria

### Participants

This scoping review is not about subjects, but rather a specific type of publication. Specifically, we will be including published reviews that call themselves a systematized review.

### Concept

The methodological elements or characteristics found in evidence synthesis reviews, as selected from the reporting guideline PRISMA (2020) [9] or a critical appraisal tool, AMSTAR-2 [4]. Specifically, we will be looking at elements relating to the search, selection, data extraction, and risk of bias steps of the review process, as well as other elements of quality such as number of authors, and involvement of other specialists, such as librarians.

### Context

Global. We will not limit to publications from any specific region.

### Types of sources

Published systematized reviews. To be considered a systematized review, the articles must specifically include the words systematized (or systematised) and review within the title of the

publication, and describe the methodology used to collocate and synthesize existing literature for their review.

## Methods

The proposed scoping review will be conducted in accordance with the JBI methodology for scoping reviews [10].

The protocol is being reported according to the items of PRISMA-P reporting guidelines that are relevant to a scoping review [11].

### Search strategy

The search strategy will aim to locate only published systematized reviews that are explicitly titled as such, in accordance with our research objective to examine what is accepted as a "systematized review" by review authors, editors, and peer reviewers.

An initial limited search of Google Scholar and Web of Science Core Collection was undertaken to identify articles on the topic. The text words contained in the titles and abstracts of relevant articles, and the index terms used to describe the articles were used to develop the full search strategy. During this exploratory phase, it came to light that authors sometimes titled their reviews as a systematic review, and in the abstract used the term systematized review. In order to avoid mixing the results of systematic reviews and systematized reviews, we decided that only articles titled a systematized review would be eligible for inclusion in this scoping review.

To obtain the broadest representation possible, the information sources for this review will include Web of Science Core Collection (specifically, Science Citation Index, Social Sciences Citation Index, Arts & Humanities Citation Index, and Emerging Sources Index), Scopus (Elsevier platform), MEDLINE (Ovid platform), Google Scholar (searched via Publish or Perish), and Lens.org.

The search strings to be used in each information source are shown in Table 1 below.

The searches will not be limited by language filters. However, due to the inability of the author team to assess research in other languages, articles in a language other than English that clearly meet the inclusion criteria during screening will be collated and a citation list will be included in the supplemental material alongside the scoping review. This will allow other researchers to quickly locate these articles should they desire. The authors acknowledge that exclusively using English language search terms will fail to retrieve research for which the title is not presented in English.

No date limit will be used during the searching or screening. However, to ensure a feasible sample size for extraction and analysis, if the final number of included studies is substantially higher than 200, then studies will be retained starting from the most recent until the sample

**Table 1. Search queries designed for each information source.**

| Source | Search query |
| --- | --- |
| Web of Science Collection | ( ("systemati?ed") NEAR/3 ("review") ) (Title) |
| Scopus (Elsevier) | TITLE (("systematized" OR "systematised") W/3 ("review")) |
| MEDLINE (Ovid) | (("systemati?ed") adj4 (review)).ti |
| Google Scholar | allintitle:("systematized review" OR "systematized * review" OR "systematized * * review" OR "systematized * * * review" OR "systematised review" OR "systematised * review" OR "systematised * * review" OR "systematised * * * review") |
| Lens.org | Title: ("systematized review" ~3) OR Title: ("systematised review" ~3) |

size of 200 is reached. At that point, all studies from the same publication year as study #200 will also be included to ensure sampling from that entire year.

Results from the searches will be downloaded as RIS files, and uploaded into Covidence for deduplication and screening. Following a pilot test of 25 records, titles and abstracts will be screened by two independent reviewers for assessment against the inclusion criteria for the review. Potentially relevant sources will be retrieved in full, and selected citations will be assessed in detail against the inclusion criteria by two independent reviewers. Reasons for exclusion determined during full text screening, will be recorded and reported. Any disagreements that arise between the reviewers at each stage of the selection process will be resolved through discussion and consensus. The results of the search and the study inclusion process will be reported in full in the final scoping review and presented in a flow diagram.

## Data extraction

Data will be extracted from the included reviews by two independent reviewers using a data extraction tool developed a priori. The data extracted will include specific details of the methods described in the included studies, drawn in part from some applicable criteria in the AMSTAR-2 critical appraisal tool for systematic reviews [4] and some items from the PRISMA 2020 reporting guideline [9]. We chose elements from PRISMA 2020, a reporting standard for systematic reviews, and AMSTAR-2, a critical appraisal tool for systematic reviews, because of Grant and Booth's [6] description of the systematized review as having "elements of systematic review process while stopping short of a systematic review" [4, p.95]. This implies that the frame of reference in terms of the evidence synthesis review type that is closest to the systematized review is that of a systematic review (rather than any other type of evidence synthesis). General data items include basic details such as the discipline the review was published in, the number of authors, and the publication year. Data items related to the search step include, the number of databases, the presentation of a reproducible search strategy, date limits used, justification for limits, inclusion of grey literature, and inclusion of citation searching. Data items related to other steps of the review include, whether dual screening was conducted, whether dual data extraction was conducted, whether risk of bias was conducted, and whether a characteristics of included studies table was included in the review. Additional data items that will be extracted relate to the justification for choosing the systematized review methodology, the mention of either evidence synthesis conducting guides or reporting standards, and involvement of an information specialist.

A full list of the data categories to be extracted can be found in the draft extraction form (provided in S1 Appendix). The draft data extraction tool has been pilot tested with 2 studies. However, during the conduct of the study, the reviewers will check-in after a larger set of studies has been extracted (20 studies) to discuss if any further modifications are required, and if so, the template will be modified and revised as necessary before proceeding further. Modifications made, if any, will be detailed in the final scoping review manuscript. Any disagreements that arise between the reviewers will be resolved through discussion.

## Data analysis and presentation

The complete data extracted from each of the studies will be made publicly available, in lieu of the characteristics of included studies table, due to the large anticipated sample size (greater than 200).

The following analyses are planned using simple aggregation or basic statistical techniques (counts, averages, means, etc.).

1. The number of systematized reviews published across time and across disciplines.

2. Common characteristics of the included studies: number of databases searched, number of authors.

3. Additional methodological characteristics: the number or percentage of studies that included specific methodological characteristics (inclusion of reproducible search strategy, mention of a protocol, grey literature, citation chaining, dual screening, inclusion of risk of bias).

4. Analysis/Number of studies that provide justification for categorization of the justifications for selecting the systematized methodology, and categorization of the justifications into the following broad categories: relate to resource constraints, relate to time constraints, other.

5. Guidance documents cited.

We have not planned to analyze the following data extraction variables (date limits used, date ranges for searches, inclusion of a search date, and librarian involvement), however, we may choose to add these into the analysis if they show interesting trends.

## Supporting information

**S1 Checklist. PRISMA-P 2015 checklist.**
(DOCX)

**S1 Appendix. Data extraction instrument built in Google sheets.**
(DOCX)

## Author Contributions

**Conceptualization:** Zahra Premji, Leyla Cabugos.

**Data curation:** Zahra Premji, Leyla Cabugos.

**Formal analysis:** Zahra Premji, Leyla Cabugos.

**Funding acquisition:** Leyla Cabugos.

**Investigation:** Zahra Premji, Leyla Cabugos.

**Methodology:** Zahra Premji, Leyla Cabugos.

**Project administration:** Leyla Cabugos.

**Visualization:** Zahra Premji, Leyla Cabugos.

**Writing – original draft:** Zahra Premji, Leyla Cabugos.

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
