## [Decision Letter · Decision Letter 0]

11 Jul 2023

PONE-D-23-07632Examining the meaning and methodological characteristics of the systematized review label: A scoping review protocolPLOS ONE

Dear Dr. Cabugos,

Thank you for submitting your manuscript to PLOS ONE. After careful consideration, we feel that it has merit but does not fully meet PLOS ONE’s publication criteria as it currently stands. Therefore, we invite you to submit a revised version of the manuscript that addresses the points raised during the review process. Please submit your revised manuscript by Aug 25 2023 11:59PM. If you will need more time than this to complete your revisions, please reply to this message or contact the journal office at plosone@plos.org. Please include the following items when submitting your revised manuscript:A rebuttal letter that responds to each point raised by the academic editor and reviewer(s). You should upload this letter as a separate file labeled 'Response to Reviewers'.A marked-up copy of your manuscript that highlights changes made to the original version. You should upload this as a separate file labeled 'Revised Manuscript with Track Changes'.An unmarked version of your revised paper without tracked changes. You should upload this as a separate file labeled 'Manuscript'.

We look forward to receiving your revised manuscript.

Kind regards,

Robin Haunschild

Academic Editor

PLOS ONE

Journal Requirements:

2. Please note that in order to use the direct billing option the corresponding author must be affiliated with the chosen institute. Please either amend your manuscript to change the affiliation or corresponding author, or email us at plosone@plos.org with a request to remove this option.

3. In your cover letter, please confirm that the research you have described in your manuscript, including participant recruitment, data collection, modification, or processing, has not started and will not start until after your paper has been accepted to the journal (assuming data need to be collected or participants recruited specifically for your study). In order to proceed with your submission, you must provide confirmation.

Reviewers' comments:

Reviewer's Responses to Questions

**Comments to the Author**

1. Does the manuscript provide a valid rationale for the proposed study, with clearly identified and justified research questions?

Reviewer #1: Yes

Reviewer #2: Yes

2. Is the protocol technically sound and planned in a manner that will lead to a meaningful outcome and allow testing the stated hypotheses?

Reviewer #1: Yes

Reviewer #2: Yes

3. Is the methodology feasible and described in sufficient detail to allow the work to be replicable?

Reviewer #1: Yes

Reviewer #2: Yes

4. Have the authors described where all data underlying the findings will be made available when the study is complete?

Reviewer #1: Yes

Reviewer #2: Yes

5. Is the manuscript presented in an intelligible fashion and written in standard English?

Reviewer #1: Yes

Reviewer #2: Yes

6. Review Comments to the Author

You may also provide optional suggestions and comments to authors that they might find helpful in planning their study.

Reviewer #1: Thank you for the opportunity to review this protocol. I am looking forward to reading more about this study. I have only a few small comments.

1. In some places, the abbreviation ES is used as a singular, and in others as a plural, leaving the reader to infer. I would prefer to see it spelled out or used consistently as a plural ("evidence syntheses").

2. In line 58 the font changes from Arial to Calibri. It seems to go back and forth again later in the manuscript.

3. In line 81 there is an extra space before the punctuation.

4. I think MEDLINE is always capitalized. Scopus is not capitalized (based on vendor/database representation).

5. I'm curious why the authors chose AMSTAR-2 which is for systematic, but not necessarily systematized, reviews. Could the authors elaborate or provide justification to the reader that it is still going to be applicable and practical in this scenario? (I understand that it will -- I think a sentence to provide rationale for this decision will be helpful).

6. Lines 193... the numbered list is missing a period on item 1 (items 2 and 3 have a period). Instead of "etc." here, I would really like to see the authors list the planned data points. "Etc." makes it sound like they'll figure it out when they have the data, but I think it would be reasonable to know what data is going to be extracted a priori. Or -- I suppose this might be covered where the authors mentioned that they may contact statisticians, so do with this suggestion what you will.

7. Line 193 onward through the numbered list -- I would think it would be the "following analyses" are planned based on the list. Item #1 implies that the authors already have this analysis -- wouldn't it be more pertinent to describe it as "the number of" rather than the "growth of" -- because technically we do not know yet.

Reviewer #2: While the topic is very interesting, however, this is just a review protocol. I would be more interested to see the results. The protocol seems ok, although it needs some edits. However, my main issue is that I would hesitate to title the methodology of the current paper as “scoping review”. Scoping review goes to a more in-depth analysis of qualitative aspects of the included papers. This paper is more descriptive to be viewed as scoping review. If the authors were aimed to characterize the meaning of systematized review, then they could have searched and analyzed methodological papers related to conceptualizing systematized reviews.

Other comments:

- The authors considered systematized review as a methodology that is used for evidence syntheses. However, that is systematic reviews which are commonly used for evidence syntheses. Systematized reviews are less rigor for evidence syntheses.

- The manuscript needs some editing as it is hard to read in some cases.

- In Participants section, there is a need to give more details about the papers searched and included in the study. Same as in the context section, there is a need to give details about the databases searched.

- Why to choose PRISMA or AMSTAR methodological elements for systematized review while they are originally for systematic reviews?

7. PLOS authors have the option to publish the peer review history of their article (what does this mean?). If published, this will include your full peer review and any attached files.

Reviewer #1: **Yes: **Carrie Price

Reviewer #2: No

---

## [Author Response · Author response to Decision Letter 0]

7 Aug 2023

Reviewer 1 

Comment 1: In some places, the abbreviation ES is used as a singular, and in others as a plural, leaving the reader to infer. I would prefer to see it spelled out or used consistently as a plural ("evidence syntheses").

Authors’ response: Thank you for this comment. We agree and have fixed the inconsistent use of ES singular and plural.

Changes made to the manuscript: We have now spelled out ES in each case so there is no question as to whether we meant the singular or plural in that instance.

Comment: 2. In line 58 the font changes from Arial to Calibri. It seems to go back and forth again later in the manuscript.

Authors’ response: Thank you for noticing the inconsistency in font types. We have fixed this and used Arial 11 point font throughout the manuscript.

Changes made to the manuscript: Font type changed to be consistent across the entire protocol.

Comment3: In line 81 there is an extra space before the punctuation.

Authors’ response: We thank the reviewer for noticing this and have fixed it.

Changes made to the manuscript: Removed extra space.

Comment 4: I think MEDLINE is always capitalized. Scopus is not capitalized (based on vendor/database representation).

Authors’ response: We thank the reviewer for providing the correct capitalization of these database names.

Changes made to the manuscript: Medline changed to MEDLINE, and SCOPUS changed to Scopus in 2 sections of the protocol.

Comment 5. I'm curious why the authors chose AMSTAR-2 which is for systematic, but not necessarily systematized, reviews. Could the authors elaborate or provide justification to the reader that it is still going to be applicable and practical in this scenario? (I understand that it will -- I think a sentence to provide rationale for this decision will be helpful).

Authors’ Response: Thank you for reminding us to justify the choice of using elements from AMSTAR-2 for our readers. As the reviewer likely knows, our observation is that in using the term “systematized review” authors are signaling that they intended to use systematic review methodology as a frame of reference, and make certain departures from it. Therefore, we will use the AMSTAR 2 framework used to evaluate systematic reviews as a way to select salient attributes to examine in systematized reviews. 

Changes made to the manuscript: We added a few sentences in the data extraction section providing justification for the choice of tools from which we selected the elements of methodology to extract from each of our included studies.

Comment 6. Lines 193... the numbered list is missing a period on item 1 (items 2 and 3 have a period). Instead of "etc." here, I would really like to see the authors list the planned data points. "Etc." makes it sound like they'll figure it out when they have the data, but I think it would be reasonable to know what data is going to be extracted a priori. Or -- I suppose this might be covered where the authors mentioned that they may contact statisticians, so do with this suggestion what you will.

Authors’ response: We thank the reviewer for their comment and prompt for us to be more specific, given that this is a registered report protocol. We have added the missing periods. We have also removed the etc, and explicitly mentioned the variables we will analyze for trends (using counts, percentages, simple aggregate statistics). Furthermore, we included a statement below the list of planned analyses highlighting additional variables that are listed in our planned data extraction where analysis is not planned at this time. If the data is found to be interesting, we will include analysis of these variables and specifically mention this addition in the final manuscript. We do not intend to analyze any additional variables not listed in this protocol.

Changes made to the manuscript: We have added some additional detail to the bulleted list in the Data analysis and presentation section, expanding the list from 3 to 5.

Comment 7. Line 193 onward through the numbered list -- I would think it would be the "following analyses" are planned based on the list. Item #1 implies that the authors already have this analysis -- wouldn't it be more pertinent to describe it as "the number of" rather than the "growth of" -- because technically we do not know yet.

Authors’ response: We thank the reviewer for this gentle reminder, and the point is well taken. We do not know the direction of the trend, and have therefore used the reviewer’s suggested wording change.

Changes made to the manuscript: Changed analysis to plural, as suggested. Changed “growth” to “number” as suggested. 

Reviewer 2

Comment 1: While the topic is very interesting, however, this is just a review protocol. I would be more interested to see the results. 

Authors’ response: We thank the reviewer for their enthusiasm for our proposed topic. We, too, are eager to get started with this research in order to see the findings. For transparency and research integrity, we chose to submit our research protocol to PLOS One’s Registered Reports track, in which the first stage peer review is conducted on our research plan before the study begins. In keeping with this process, PLOS One asks us to affirm that the research described in our protocol has not started and will not start until after our paper has been accepted to the journal. We are hopeful for an acceptance in the near future so we can proceed with this research.

Changes made to the manuscript: None.

Comment 2. The protocol seems ok, although it needs some edits.

Authors’ response: Given the lack of a similar comment from reviewer 1, and no further detail, we are not sure which sections to focus on for editing. We have made all of the editing suggestions that Reviewer 1 suggested, and would be happy to do the same if Reviewer 2 would kindly point us in the right direction.

Changes made to the manuscript: None. 

Comment 3. However, my main issue is that I would hesitate to title the methodology of the current paper as “scoping review”. Scoping review goes to a more in-depth analysis of qualitative aspects of the included papers. This paper is more descriptive to be viewed as scoping review

Authors’ response: We thank the reviewer for their comment. We chose the scoping review label because according to Pollock et al (2023), “Scoping reviews are descriptive in nature; they aim to map the available evidence or identify characteristics or factors. For the most part, there will be no need for scoping review authors to go beyond basic descriptive analysis.“ They further state, “Most scoping reviews will analyze data items by quantifying text and doing frequency counts of data extraction items.“ Given the match between our proposed research plan and the above description from guidance published by JBI, which sets standards in evidence synthesis methodology, we felt it appropriate to call our review a scoping review. 

Pollock, D., Peters, M. D. J., Khalil, H., McInerney, P., Alexander, L., Tricco, A. C., Evans, C., de Moraes, É. B., Godfrey, C. M., Pieper, D., Saran, A., Stern, C., & Munn, Z. (2023). Recommendations for the extraction, analysis, and presentation of results in scoping reviews. JBI Evidence Synthesis, 21(3), 520. https://doi.org/10.11124/JBIES-22-00123

Changes made to the manuscript: None. 

Comment 4. If the authors were aimed to characterize the meaning of systematized review, then they could have searched and analyzed methodological papers related to conceptualizing systematized reviews.

Authors’ response: We thank the reviewer for their comment. Their comment aligns with our motivation. We had considered designing the study as a methodology review, which according to the Cochrane Methodology Review Group would “cover studies assessing the methodology of research… rather than the effects of the care [Cochrane is focused on medical interventions but the guidance is applied in other disciplines] itself” (Clarke et a., 2011). However, in our exploratory searching, we could not find any existing methodological research on systematized reviews, hence we would not have a suitable population for a methodology review. Thus, in order to determine the extent and common characteristics of published systematized reviews at this early stage in the research community’s consideration of the systematized review phenomenon, we must analyze the systematized reviews themselves, and will use descriptive measures as expected for a scoping review.

Clarke, M., Oxman, A. D., Paulsen, E., Higgins, J. P. T., & Green, S. (2011). Appendix A: Guide to the contents of a Cochrane Methodology protocol and review. Cochrane Handbook for systematic reviews of interventions.

Changes made to the manuscript: None. We justified our choice to analyze the systematized reviews in the paragraph immediately above the review question, and we believe this should be sufficient.

Comment 5. The authors considered systematized review as a methodology that is used for evidence syntheses. However, that is systematic reviews which are commonly used for evidence syntheses. Systematized reviews are less rigor for evidence syntheses.

Authors’ response: According to Grant & Booth (2009) and Sutton et al (2019), systematized reviews are a type of evidence synthesis, different from the other forms of evidence synthesis, and expected to be conducted with less rigor, which is sometimes due to some type of resource restraint (single author, less time, few resources). The premise of our study explicitly addresses the concern that systematized reviews may be less rigorous than systematic reviews. However, since systematized reviews are being published and may potentially be used for decision-making, and since there is no “formal” methodology for this review type, we believe there is value in seeking to characterize its methodological elements to provide a picture of the state of the methodology as it exists to date.

Changes made to the manuscript: None. 

Comment 6. The manuscript needs some editing as it is hard to read in some cases.

Authors’ response: To address this concern, we will need the reviewer to identify illustrative examples and comment on what makes them hard to read. We welcome any additional details so we can improve the manuscript's readability.

Changes made to the manuscript: None. 

Comment 7. In the Participants section, there is a need to give more details about the papers searched and included in the study. Same as in the context section, there is a need to give details about the databases searched.

Authors’ response: We thank the reviewer for their comment. Once the study has been conducted, we will follow the reporting guidelines in the PRISMA extension for scoping reviews to document the numbers of sources of evidence screened, assessed for eligibility, and included in the review in the results of our final manuscript. 

In describing our eligibility criteria, we chose to describe the details of the types of included studies in the section below the PCC called Types of Sources. However, we have now reiterated the type of included study in the participants section as well, based on the reviewers comments. We believe that the existing description of the Context as global is fitting and sufficient as we are not limiting to publications from any specific region. We understand the reviewer’s point about stating the database details in this section, but we think that the list of sources is better suited to the search methods section of a scoping review protocol.

Changes made to the manuscript: Added a sentence in the Participants section stating that our included studies are published systematized reviews.

---

## [Decision Letter · Decision Letter 1]

23 Aug 2023

Examining the meaning and methodological characteristics of the systematized review label: A scoping review protocol

PONE-D-23-07632R1

Dear Dr. Cabugos,

We’re pleased to inform you that your manuscript has been judged scientifically suitable for publication and will be formally accepted for publication once it meets all outstanding technical requirements.

Kind regards,

Robin Haunschild

Academic Editor

PLOS ONE

Additional Editor Comments (optional):

Reviewers' comments:

Reviewer's Responses to Questions

**Comments to the Author**

1. Does the manuscript provide a valid rationale for the proposed study, with clearly identified and justified research questions?

Reviewer #1: Yes

Reviewer #2: Yes

2. Is the protocol technically sound and planned in a manner that will lead to a meaningful outcome and allow testing the stated hypotheses?

Reviewer #1: Yes

Reviewer #2: Yes

3. Is the methodology feasible and described in sufficient detail to allow the work to be replicable?

Reviewer #1: Yes

Reviewer #2: Yes

4. Have the authors described where all data underlying the findings will be made available when the study is complete?

Reviewer #1: Yes

Reviewer #2: Yes

5. Is the manuscript presented in an intelligible fashion and written in standard English?

Reviewer #1: Yes

Reviewer #2: Yes

6. Review Comments to the Author

You may also provide optional suggestions and comments to authors that they might find helpful in planning their study.

Reviewer #1: The authors took great care to respond to my comments, provide justification, and specify if and where changes were made to the manuscript. I feel that the authors have recognized and addressed my comments. I am excited for what this paper will add to the corpus of literature, since not very much has been written about systematized reviews. This work is greatly needed. I thank you for the opportunity to review this work and suggest unequivocally that it is accepted for publication.

Reviewer #2: As for my previous comments, I expect th eauthors justify well in the protocol their use of the scoping review protocol. And, if it is possible give the maniscript for professional eding service.

7. PLOS authors have the option to publish the peer review history of their article (what does this mean?). If published, this will include your full peer review and any attached files.

Reviewer #1: **Yes: **Carrie Price

Reviewer #2: **Yes: **Sirous Panahi

---

## [Editor Report · Acceptance letter]

29 Aug 2023

PONE-D-23-07632R1 

Examining the meaning and methodological characteristics of the systematized review label: A scoping review protocol 

Dear Dr. Cabugos:

I'm pleased to inform you that your manuscript has been deemed suitable for publication in PLOS ONE. Congratulations! Your manuscript is now with our production department. 

Kind regards, 

on behalf of

Dr. Robin Haunschild 

Academic Editor

PLOS ONE